# Incorporating human mobility to enhance epidemic response and estimate real-time reproduction numbers

Mousumi Roy[1]*, Hannah E. Clapham[1], Swapnil Mishra[1,2,3,4]*

**1** Saw Swee Hock School of Public Health, National University of Singapore, Singapore, **2** Institute of Data Science, National University of Singapore, Singapore, **3** Department of Statistics and Data Science, National University of Singapore, Singapore, **4** Research Office, National Centre for Infectious Diseases, Singapore

* mousumiroy1991@gmail.com (MR); swapnil.mishra@nus.edu.sg (SM)

## Abstract

Human mobility plays a critical role in the transmission dynamics of infectious diseases, influencing both their spread and the effectiveness of control measures. In the process of quantifying the real-time situation of an epidemic, the instantaneous reproduction number $R_t$ appears to be one of the useful metrics widely used by public health researchers, officials, and policy makers. Since individuals can contract infections both within their region of origin and in other regions they visit, ignoring human mobility in the estimation process overlooks its impact on transmission dynamics and can lead to biased estimates of $R_t$, potentially misrepresenting the true epidemic situation. Our study explicitly integrates human mobility into a renewal-equation based disease transmission model to capture the mobility-driven effect on transmission. By incorporating pathogen-specific generation-time distribution, observational delay, the framework is epidemiologically informed and flexible to a wide range of diseases. We primarily validate the approach using simulated data, and demonstrate the limitations of estimating $R_t$ without considering mobility. We then apply it to two real-world mobility settings using SARS-CoV-2 mortality data: the regions of England and the LTLAs of North East region of England, and uncover the mobility driven effect on transmission at different spatial resolutions. This framework uses non-identifiable and widely accessible publicly available datasets, demonstrating its practical applicability and supporting better-informed and more targeted public health measures.

## Author summary

The real-time or instantaneous reproduction number $R_t$ is a key metric for assessing the state of an epidemic at any given time. When estimating these numbers across multiple connected regions, human mobility plays a crucial role, as movement patterns significantly influence disease transmission. Traditional

**Data availability statement:** All the codes and data related to this study are available at https://github.com/mlgh-sg/Spatial_transmission and https://doi.org/10.5281/zenodo.17077496.

**Funding:** SM and MR acknowledge support from National Research Foundation, Singapore, under its NRF FELLOWSHIP (NRF-NRFF15 - 2023 - 0010) awarded to SM. The funders had no role in study design, data collection and analysis, decision to publish, or preparation of the manuscript.

**Competing interests:** The authors have declared that no competing interests exist.

epidemic models often assume homogeneous mixing, which does not reflect real-world interactions. On the other hand, individual-based models incorporate heterogeneous mixing at individual level but demands an extremely refined data and substantial computational support. To address these challenges, we employ a renewal equation-based transmission framework, particularly useful for its effectiveness in real-time epidemic analysis, by incorporating heterogeneous mobility flows at a chosen spatial resolution. This yields the estimates of spatially connected instantaneous reproduction number for each region. This improved understanding enables better assessment of the impact of mobility on disease transmission spread, and provides valuable insights for designing targeted epidemic control and intervention strategies.

## 1 Introduction

Human mobility significantly influences the transmission dynamics of infectious diseases [1,2]. Hence, its been a constant effort from the researchers to incorporate human mobility within a transmission model in the process of analyzing disease spread [3–5]. While long-range mobility like air-traffic flow between countries quantifies the importation risks for a country, daily commuting travel structure builds the onward transmission routes within a country after initial importation [6–8]. By daily commuting flows, people move across different regions and increase contact opportunities at distant places, that enables inter-regional disease spread. In contrast, during COVID-19 pandemic, lockdowns and other non-pharmaceutical interventions (NPIs) led to substantial reductions in mobility within any country, and accordingly disease transmission [9,10]. This highlights the need to account for mobility in the real-time estimation of reproduction numbers, and therefore, ignoring mobility in transmission models can lead to biased parameter estimates and compromise the accuracy of future predictions. In this study, we use *human mobility* to describe the movement of people between regions, which creates *spatial connectivity* and leads to a *mobile population* that affects how diseases spread. These three terms *human mobility*, *spatial connectivity*, and *mobile population* are used interchangeably throughout the paper to convey a similar meaning.

Various modeling strategies have been implemented to study infectious diseases at the population level, each presenting unique benefits and drawbacks. Compartmental mechanistic models [11], in the absence of a spatial component and more advanced mixing assumptions (e.g., contact matrices) may lead to biased estimation due to their assumption of homogeneous mixing, a condition that rarely reflects the real-world mixing pattern [12,13]. This leads to questions about the reliability of the model within a heterogeneous setting. A network-based model [14,15], which overlays a network on top of compartmental mechanistic models, can incorporate information about heterogeneous mixing. In this context, Metapopulation model partitions the total population into subpopulations, with mobility between them [4,16], often used to understand the mechanism and study real-time transmission in a spatially

explicit setting [17,18]. However it is associated with several challenges, such as incorporating stochastic extinction and reintroduction of the disease [19]. On the other hand, individual-level models [20,21] capture the nuanced complexities of human mobility and individual heterogeneity, but are computationally intensive and require granular data for validation. In practice, studies often use proxy-mobility datasets, such as, mobile-phone, public-transport records [22,23], or substitute established mobility models when empirical data are unavailable [24–26]. An alternative transmission model is the approach based on the renewal equation [27,28], which effectively models infection at the population level and is particularly valuable for real-time analysis of disease transmission. Incorporating a spatial component into the renewal equation-based framework allows to model daily commuting flow within a population and greatly enhance the estimation of spatially connected parameters on real-time. This improves the understanding of real-time transmission dynamics in the context of a population with heterogeneous mobility.

In the process of modeling and understanding a real time epidemic situation, the instantaneous or real-time reproduction number ($R_t$) serves as a metric that quantifies the average number of secondary cases generated from a primary infection at any particular time point. This metric is crucial for assessing the current state of the epidemic and evaluating the effectiveness of any intervention strategies implemented throughout its course [29,30]. Inaccurate estimation of $R_t$ can lead to a misleading understanding of the epidemic situation, potentially resulting in misallocation of resources and improper implementation of control policies. To estimate the reproduction number $R_t$, various forms of observational data, such as, cases, hospitalizations, deaths, etc. can be utilized [28,31]. Several estimation methods are available, each with its distinct advantages and methodological limitations [27,31,32]. When building the modeling framework, it is essential to consider the delay between the actual infection and the reporting of observational data that are used for estimation, particularly towards the end of the timeline of the datasets [33]. When daily observational data are lacking, a temporally aggregated dataset can also be used for estimation purposes [34,35].

Despite significant efforts by infectious disease modelers to integrate human mobility into transmission models in order to understand its influence on transmission pattern, [36–38], this area still requires further investigation. In contrast to previous literature with renewal-equation based framework where in most of the studies mobility reduction appears as an aspect of behavioral changes [28,39], we use explicit human mobility flow within the model to account for inter-regional mobility effect over the transmission dynamics. We apply the framework to the initial phase of the COVID-19 pandemic in England as a case study to illustrate its real-world applicability. Consequently, this study systematically investigates the influence of commuting populations and the epidemic conditions of interconnected regions on disease transmission dynamics, using both simulated data and a case study with necessary adaptations to the model to reflect the specific context. Furthermore, we observe the spatial scale itself is a vital factor, as working with a spatial scale where the joint effects of mobility and epidemic situation are not visible may diminish the observable impact of mobility, which may be more apparent at a comparatively coarse resolution.

## 2 Materials and methods

In this study, we consider interconnected regions characterized by commuting populations, reflecting daily movement patterns from home to various destinations. We use a discrete-time renewal equation based approach [27], integrated with a spatial component, to assess the real-time impact of mobility on transmission scenario. This approach helps us understand how heterogeneous commuting patterns across different spatial scales influence transmission dynamics. New infections in any region arise not only from the infected population within the region but also from the mobile population between connected regions. It provides a thorough evaluation of how regional connectivity affects epidemic dynamics within a specific region. The estimated instantaneous reproduction number using this framework accounts both within region transmission and the effect of epidemic status of neighboring regions with the corresponding inter-regional mobility.

## 2.1 Modeling infection within a region

To model the infections within a region, we use the renewal-equation based framework [27,28,32], where new infections at time $t$ are described as,

$$I_t = (S_t/N)R_t \sum_{\tau=0}^{t-1} I_\tau g_{t-\tau} \tag{1}$$

where, $S_t = N - (\sum_{s=1}^{(t-1)} I_s)$ represents the susceptible population at time $t$, and $N$ denotes the total population. Eq 1 illustrates that the number of new infections at any given time depends on the prior infections, with their contributions weighted by the generation time distribution, $g$, which captures time interval between primary and secondary infections. Here $g$ is derived by discretizing the pathogen-specific density function $g(t)$ as $g_s = \int_{s-0.5}^{s+0.5} g(t)dt$ for $s = 2, 3, 4, \ldots$ and $g_1 = \int_0^{1.5} g(t)dt$. The instantaneous reproduction number $R_t$ quantifies the average number of secondary infections caused by a single primary infection at time $t$. Eq 1 provides a robust foundation for understanding the transmission dynamics within a single region, which we refer to as the national model.

## 2.2 Extension of the framework with a spatial component

Central to this study is the advancement of the existing epidemiological framework through the integration of a spatially explicit component, which systematically incorporates human mobility data to elucidate the mechanistic role of population movement in shaping the spatiotemporally heterogeneous disease transmission dynamics. A schematic representation of the extended framework is shown in Fig 1. For simplicity, we consider two regions: region 1 and region 2 represented by two different colors. The population in each region comprises susceptible and infected individuals, with a portion of the population commuting between regions. Additionally, we assume that recovery from the disease confers lifetime immunity, as recovered individuals cannot be reinfected. The mobility between two regions $m_{ij}(j \rightarrow i)$, represents the proportion of the population of region $j$ that commutes to region $i$. Hence, in Fig 1, $m_{12}$ denotes the fraction of the population of region 2 that commutes to region 1, and similarly, $m_{21}$ represents the reverse flow. The diagonal elements of the mobility matrix $m_{ii}$ indicate the fraction of the population that does not commute, staying within their respective regions. The presence of mobile population implies that the infection in any region are influenced not only by local transmission but also by interactions with infected individuals from other connected regions. To illustrate the transmission dynamics in more detail, we focus on infection pathways affecting the population of region 1. As illustrated in Fig 1, the red ovals indicate three different routes through which individuals in region 1 can acquire infection. Susceptibles in region 1 can be exposed to infected individuals within their own region or can acquire infection by visiting region 2 and contract infection there. In summary,

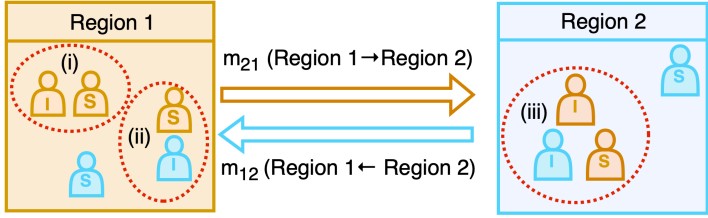

**Fig 1**. **Schematic representation of disease transmission dynamics with mobility.** The diagram illustrates two interconnected regions (Region 1 and Region 2), where individuals are classified as susceptible (S) and infected (I) can move between regions. The colors indicate the region of origin for each individual. The arrows represent mobility flows: $m_{21}$ (movement from Region 1 to Region 2) and $m_{12}$ (movement from Region 2 to Region 1). The diagram describes the disease transmission pathways for region 1, highlighted by red ovals: (i) local transmission due to interactions between susceptible and infected individuals of region 1 within region 1, (ii) infected individuals of region 2 commute to region 1, can transmit infection to the susceptible population of region 1, (iii) susceptible individuals from Region 1 who travel to Region 2, acquire the infection in region 2, and return as infected individuals. This framework captures the role of human mobility in shaping spatial disease transmission patterns.

infection within any region can occur through three primary pathways: (i) local transmission within the region by its own infected population, (ii) infection through incoming infected commuters from other regions, and (iii) acquisition of infection during visit to another regions. These mobility-driven transmission mechanisms underscore the complexity of disease spread in interconnected populations.

Drawing from these observations, we develop a modeling framework that explicitly incorporates human mobility into disease transmission dynamics. Using Eq 1, as the foundational structure, we extend the model by integrating a spatial component to account for mobility flows between regions. For the modeling framework, we consider a total of $M$ interconnected regions, and we model the infection dynamics in a region denoted as $i$, with the number of infections at time $t$ represented as $I_{i,t}$, while allowing for population movement across $M$ regions. At time $t$, the susceptible individuals from region $i$ commuting to another region $j$ are represented as $m_{ji}(t)S_{t,i}$. These susceptible individuals are exposed to all the infected population in region $j$ at that time, given by $\sum_{k=1}^{M} m_{jk}(t) \sum_{\tau=0}^{t-1} I_{\tau,k}g_{t-\tau}$. This interaction between susceptible individuals from region $i$ and infected population in region $j$ results in new infection of region $i$, albeit occurring in region $j$. Since individuals from region $i$ can commute to any regions, summing over all regions $j$ yields the total infections in region $i$ that is given by,

$$I_{t,i} = \sum_{j=1}^{M} \frac{m_{ji}(t)S_{t,i}}{\left[\sum_{l=1}^{M} m_{jl}(t)N_l\right]} \cdot R_{t,j} \left[\sum_{k=1}^{M} m_{jk}(t) \sum_{\tau=0}^{t-1} I_{\tau,k}g_{t-\tau}\right], i = 1, 2, 3, ..., M. \tag{2}$$

Here, the denominator serves as a normalization term, describing the effective population in region $j$ at that time. $R_{t,j}$ is the reproduction number of region $j$, indicates the average number of secondary infections occurring from a primary infection in that region. $g$ is the generation time distribution, and $N_i$ is the total population of region $i$, while $S_{t,i}$ represents the number of susceptible individuals left in region $i$ at time $t$. In the special case of completely isolated regions, the mobility matrix becomes an identity matrix (unity on the diagonal and with zero elsewhere). With this simplification, in the absence of inter-regional mobility, Eq 2 reduces to Eq 1 for each region. In the context of the framework's time scale, if individuals do not spend an entire day in another region, the framework is flexible enough to adopt that fact. A detailed simulation is provided in S12 Fig. However, in this work, we consider a daily time scale throughout the study, based on the available data.

As the next step, for each region $i$, the model is fitted to available observational data, such as infection data, case counts, or death data, to estimate the region-specific instantaneous reproduction number $R_t$. The model can be adjusted based on the nature of the available datasets and the specific context. We demonstrate the utility of our model through two distinct applications. First, we employ simulated daily incidence data as observations to validate the model in a controlled setting. Second, we apply the model to real-world data by analyzing the early spread of SARS-CoV-2 in England, using weekly mortality data as observations. These examples illustrate the flexibility of the framework in accommodating both different types of data and their aggregated nature. This spatially extended framework offers a comprehensive understanding of the impact of human mobility on disease transmission dynamics across interconnected regions. By explicitly incorporating mobility flows into the infection process, the model captures both local and mobility-driven transmission, providing a more realistic representation of pathogen spread in spatially structured populations. Here, we provide a detailed description of three sources of infection.

## 2.3 Sources of infections

Infections within a region can originate from three different sources. Using Eq 2, we segregate the infected population based on the three different sources of transmission:

1. **Infections driven by own infected population within the region:** These infections occur within the region when susceptible individuals contract the disease from the infected population of their own region. This category captures

purely local transmission dynamics and can be expressed as:

$$I_{t,i}^{in,loc} = \frac{m_{ii}(t)S_{t,i}}{\left[\sum_{l=1}^{M} m_{il}N_l\right]} \cdot R_{t,i}\left[m_{ii}(t)\sum_{\tau=0}^{t-1} I_{\tau,i}g_{t-\tau}\right], i = 1,2,3,...,M. \tag{3}$$

2. **Mobility induced infections within the region:** These infections are caused by infected individuals from other regions visiting the region and transmitting the disease to the local susceptible population. This category is defined as infections due to mobility inside the region and can be expressed as,

$$I_{t,i}^{in,mob} = \frac{m_{ii}(t)S_{t,i}}{\left[\sum_{l=1}^{M} m_{il}N_l\right]} \cdot R_{t,i}\left[\sum_{k=1,k\neq i,}^{M} m_{ik}(t)\sum_{\tau=0}^{t-1} I_{\tau,k}g_{t-\tau}\right], i = 1,2,3,...,M. \tag{4}$$

3. **Mobility induced infections outside the region:** These infections occur when individuals from a region commute to other regions, become exposed there, and get infected. This category is defined as infections due to mobility outside the region and can be expressed as :

$$I_{t,i}^{out,mob} = \sum_{j=1,j\neq i}^{M} \frac{m_{ji}(t)S_{t,i}}{\left[\sum_{l=1}^{M} m_{jl}N_l\right]} \cdot R_{t,j}\left[\sum_{k=1}^{M} m_{jk}(t)\sum_{\tau=0}^{t-1} I_{\tau,k}g_{t-\tau}\right], i = 1,2,3,...,M. \tag{5}$$

## 2.4 Simulated data generation

We use a simulated infection dataset to validate the framework. The simulated dataset assumes three regions with total populations of 20,000,000, 10,000,000, and 15,000,000, respectively. These regions are interconnected through a mobility matrix (see Fig 2b), where each column corresponds to the outflow of individuals from a specific region (i.e., the $j$th column represents outflows from region $j$), while each row represents the inflows into a region (i.e., the $i$th row represents inflows into region $i$). Consequently, the sum of each column equals unity. According to the intrinsic characteristic of the framework, infections are assumed to occur homogeneously within each region governed by the corresponding reproduction numbers, $R_{t,i}, i = 1,2,3$ (see Fig 2c). We randomly choose three different countries (IND, GRB, ITA) from the repository [40], and take the time series of $R_t$s for a time period of 350 days. Based on the reproduction numbers, mobility matrix, and using Eq 2, simulated data are generated as shown in Fig 2d. The colors of the reproduction number and incidence data correspond to their respective regions. The generation time distribution is obtained by discretizing the density function $g \sim Gamma(6.5, 0.62)$ [28,41], a SARS-CoV-2 specific generation time distribution. Here, we consider the gamma distribution with mean 6.5 and CV 0.62. However, in real-world scenarios, human mobility, total population, and observational datasets are typically the only available data sources. Using these, the goal is to estimate the instantaneous reproduction numbers $R_t$, for all regions to understand the epidemic dynamics in the presence of human mobility.

## 2.5 Model adjustments and assumptions for estimating $R_t$ with the simulated data

We define $R_{t,i}$ as the effective reproduction number for the $i$th region at time $t$ and is modeled as geometric random walk, where $log(R_{t,i})$ is updated according to a normal distribution with the mean $log(R_{t-1,i})$ and a standard deviation $\sigma_i$ as follows:

$$log(R_{t,i}) \sim \mathcal{N}(log(R_{t-1,i}), \sigma_i) \tag{6}$$

Here, $R_{0,i}$ is the reproduction number at $t = 0$ defined as the average number of secondary cases coming from an infected individual at the start of the outbreak. It is assigned a weak prior $log(R_{0,i}) \sim \mathcal{N}(-0.1, 0.5)$ to consider a wide range

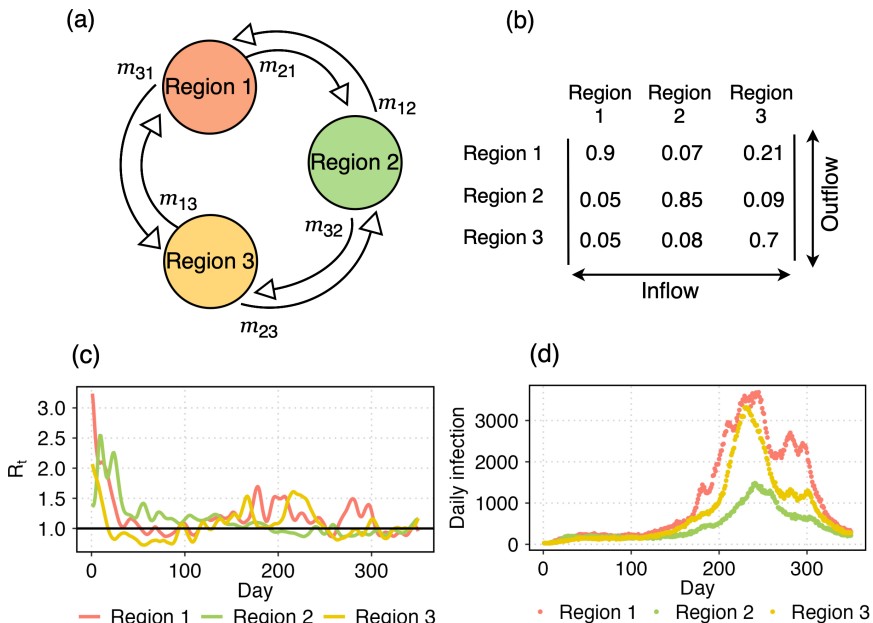

Fig 2. **Simulated dataset.** (a) Three regions with the associated mobility flow. (b) The mobility matrix where each element $m_{ij}$ denotes the fraction of population of region $i$ commutes to region $j$. Therefore, the columns show the outflows from the respective region and each row represents the inflow to that specific region. (c) Instantaneous reproduction numbers $R_t$ for the three regions. (d) With the mobility matrix and reproduction numbers, we simulate daily infection data using Eq 2.

of values. $\sigma_i \sim \mathcal{N}^+(0, 0.05)$ follows a prior of truncated normal distribution (constrained to positive values). For connected $R_t$ estimation, we use the mobility matrix as defined in Fig 2b. We use CmdStan [42] for model fitting, with NUTS sampler. 4 chains are generated in parallel for 500 iterations after 500 warm-up. There are no divergent transitions.

### 2.6 Model adjustments and assumptions for case study: SARS-CoV-2 in England and North East region of England (2020)

We apply the framework 2 to analyze the initial phases of the COVID-19 pandemic in England during 2020 as a case study. To address the challenge of low reporting rates in the early stages, we use weekly death data to estimate $R_t$ for the nine regions of England: North East, North West, Yorkshire and the Humber, East Midlands, West Midlands, East, London, South East and South West. The mobility matrix (see S3a Fig) is constructed over the nine regions using data from [43], which reports 2011 census estimates for individuals aged 16 years and over in England and Wales with their place of residence and workplace. We calculate the proportions of outflows to obtain the mobility matrix that is compatible with the model 2. During the lockdown periods, we set the mobility matrix to identity ($m_{ij} = 1$ if $i = j$, and 0 otherwise), ignoring inter-regional mobility. The population data are obtained from [44], and weekly death data [45] for each region are shown in S4 Fig. We outline below the framework used to estimate reproduction numbers.

We use the same generation time distribution, $g \sim Gamma(6.5, 0.62)$ [28,41], as in the previous section to simulate infections. For death modeling, we consider the infection-to-death delay distribution $\pi \sim Gamma(5.1, 0.86)$ $+ Gamma(17.8, 0.45)$, which combines the infection-to-onset and onset-to-death [28,39,46] delays. Here, the gamma distributions are parametrized as Gamma(mean, CV).

The weekly death data $D_{w,i}$ for region $i$ in week $w$ is modeled using a negative binomial distribution:

$$D_{w,i} \sim \text{Negative binomial}\left(d_{w,i}, d_{w,i} + \frac{d_{w,i}^2}{\psi}\right) \tag{7}$$

Here, $\psi \sim \mathcal{N}^+(0,5)$ [39]. Here, $d_{w,i}$ is the expected number of weekly deaths, calculated as the sum of daily deaths over seven days. Daily deaths $d_{t,i}$ are derived from daily infections $I_{t,i}$ using the discrete sum,

$$d_{t,i} = ifr_i^* \sum_{\tau=0}^{t-1} I_{\tau,i} \pi(t-\tau), \tag{8}$$

where $ifr_i^* = ifr \cdot N(1, 0.1)$ represent the infection fatality ratio [28] of England with a noise around it corresponding to region $i$.

To estimate the instantaneous reproduction number $R_{t,i}$ for each region, we account for temporal changes in human behavior utilizing activity trends derived from the google mobility report [9]. We use five different types of information from Google mobility data: retail and recreation, groceries and pharmacies, parks, transit stations, and workplaces. Here, we define $R_t$ as follows:

$$R_{t,i} = R_{0,i} \cdot f\left(-X_i P_t \alpha_{t,i} - \epsilon_{w_{t,i}}\right), \tag{9}$$

where, $f(x) = 2exp(x)/(1+exp(x))$, is twice the inverse logit function. $R_{0,i} \sim \mathcal{N}(3.28, \kappa)$ with $\kappa \sim \mathcal{N}^+(0, 0.5)$ for all regions [28,39]. Here, $R_{0,i}$ is the initial reproduction number for region $i$. In this context, it is the value of the reproduction number before fitting starts. $\epsilon_{w_{t,i}}$ is the weekly effect that follows a random walk centered at 0, with the week index $w_{t,i} = [\frac{t-t_0}{7}] + 1$, where $t_0$ is the first day of seeding.

$X_i$ is the region-specific effect for covariates $\alpha_{t,i}$, which are the averages of five different dimensions observed from Google mobility data. The implementation of NPIs typically leads to a significant decline in both population mobility and disease transmissibility. However, in many countries, the adoption of social distancing behaviours weakened the correlation between mobility and transmissibility following the relaxation of control measures [47–49]. To account for this effect, we introduce the parameter $P_t$, which captures behavioural changes associated with different lockdown phases, with distinct values $x$, $y$, and $z$ representing the periods up to the end of the first lockdown, from the end of the first lockdown to the end of the second lockdown, and the post-second lockdown period, respectively.

We assume that initial seeding begins six weeks prior to the date on which each region in England reaches 10 cumulative reported deaths. From that date, we seed equal initial infections for six consecutive days at each region, modeled as [39]:

$$I_{1,i} = I_{2,i} = I_{3,i} = I_{4,i} = I_{5,i} = I_{6,i} \sim \text{Exponential}(\frac{1}{\tau}), \tag{10}$$

$$\tau \sim \text{Exponential}(0.01), \quad i = 1, 2, 3, ..., M.$$

By tailoring the model to account for spatial and temporal variations in epidemic trends, the framework provides a robust tool for understanding and managing regional dynamics of SARS-CoV-2 in England in the year 2020. The model fitting is shown in S5 Fig for nine regions of England.

We then apply the framework to a finer spatial scale, focusing to the North East region, which is selected for detailed analysis because nearly 99% of its population does not commute to other regions. This characteristic makes it an ideal area for studying the relationship between human mobility and disease transmission on a more localized scale. The mobility matrix (see S3b Fig) is constructed for the twelve lower-tier local authorities (LTLAs) within the North East region based

on the same mobility data [43], ignoring mobility to regions outside of it. We use the same model described in Eq 2 and Sect 2.6, with death data specific to the North East region (see S9 Fig). The model fitting across the 12 LTLAs are illustrated in S10 Fig.

We use CmdStan [42] for model fitting, with NUTS sampler. 4 chains are generated in parallel for 500 iterations after 1000 warm-up. There are no divergent transitions.

## 3 Results

### 3.1 Model validation with simulated dataset

To validate the proposed framework (described in Sect 2), we compare the estimated $R_t$s with the ground truth (see Fig 2c) used in the simulation. This validation ensures that the framework reliably captures the transmission dynamics under controlled conditions before applying it to real-world scenarios. Fig 3a–3c presents the results for the simulated dataset, showing estimated reproduction numbers for three regions. The disconnected and connected $R_t$ estimates are shown in red and green, representing scenarios without and with inter-regional mobility, respectively. In the disconnected model, mobility-driven infections are ignored, assuming individuals remain within their respective regions and infections occur exclusively within regional boundaries. This simplification produces biased $R_t$ estimates. Table 1 shows the correlation between one-step increments in $R_t$ for the connected (green) and disconnected (red) estimates against the ground truth (blue). The consistently higher correlation between the connected $R_t$ and ground truth supports the observation that disconnected $R_t$ can lead to a severe misleading understanding of transmission patterns.

To further investigate the relationship between mobility-driven infections and overall transmission dynamics, Fig 3d–3i shows the absolute number and proportion of infections arising from three distinct sources. These figures indicate that a considerable proportion of infections is influenced by mobility. This effect becomes particularly important when $R_t$ varies across regions and the epidemic behaves differently from one region to another, an effect overlooked by the disconnected $R_t$.

Table 2 produces a quantitative support that shows a positive correlation between the residuals of disconnected $R_t$ with respect to the ground truth and the proportion of mobility-driven new infections. When the mobile population gets fewer infections in other regions than they would without commuting, disconnected $R_t$ underestimates (residual negative), and if the number of such infections is higher, it overestimates the epidemic (residual positive). For example, around day 100, the disconnected $R_t$ underestimates the true $R_t$ in Region 2. Around that time, as 15% population of region 2 commutes to regions 1 and 3, where reproduction numbers are lower than region 2, it reduces mobility driven infection outside the region, and as well as the total number of infections in region 2. However, disconnected $R_t$ fails to capture this mobility-driven lowering effect outside the region and leads to an underestimation of the true epidemic situation. On the other hand, around day = 220 in region 2, by failing to account for that mobility-driven increasing infection outside the region, disconnected $R_t$ overestimates the epidemic situation. Therefore, disconnected $R_t$ underestimates or overestimates an epidemic when mobility reduces or increases infections compared to a no-mobility scenario. Further discussion of the effects of mobility and regional epidemic conditions on transmission dynamics is provided in S1 and S2 Figs.

These results validate the effectiveness and necessity of incorporating human mobility into the model to accurately estimate reproduction numbers and better understand the epidemic situation. This approach is essentially critical in scenarios involving spatial heterogeneity in transmission dynamics.

### 3.2 Case study: Instantaneous reproduction number estimates over the regions of England

In Fig 4, we present the estimates of the connected and disconnected instantaneous reproduction number ($R_t$) (mean and 95% credible interval) for the nine regions of England, comparing results with and without account for mobility. The vertical dotted lines indicate the start and end of the first and second lockdowns. We observe that the connected and

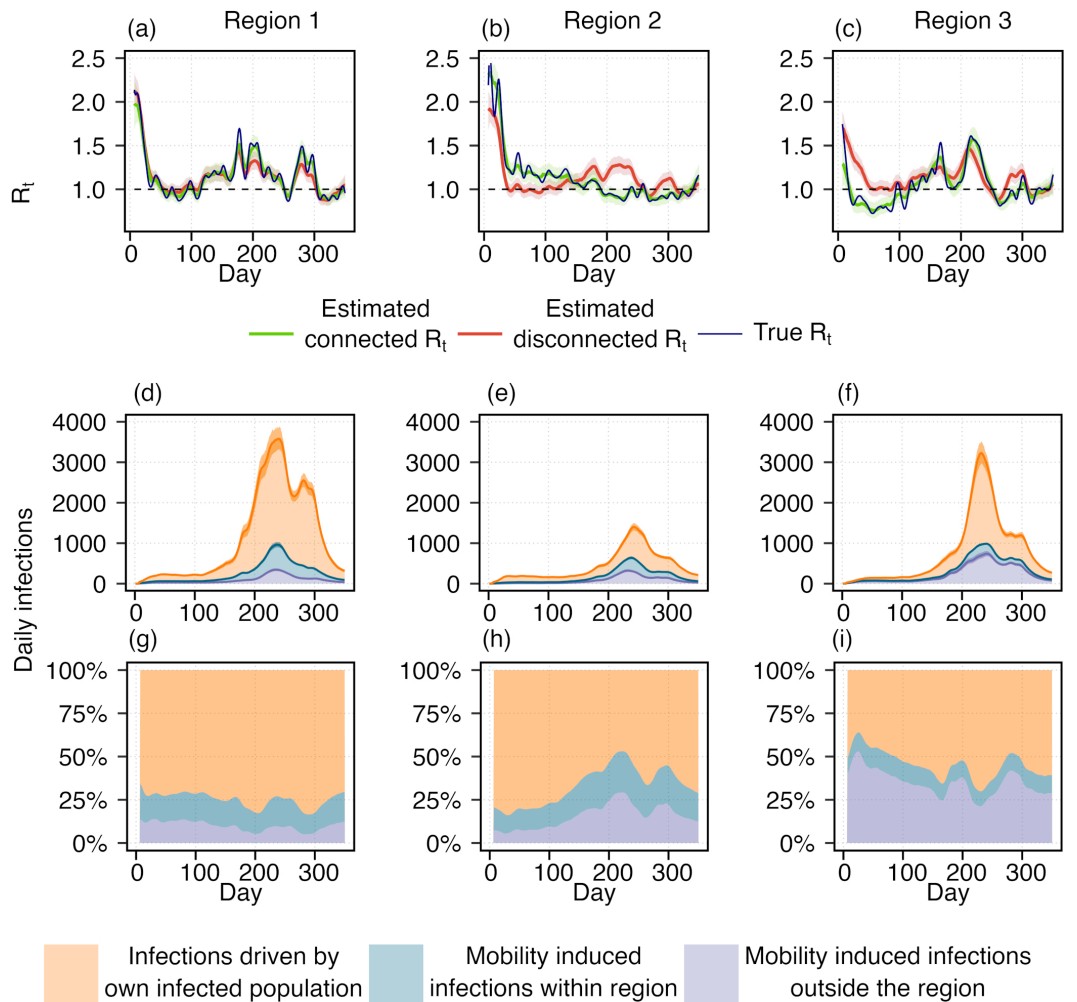

**Fig 3**. **Estimation of time-varying reproduction number ($R_t$) using simulated data and contribution of mobility-driven infections.** Top row (a–c): Blue curves represent the true $R_t$ trajectories for three regions, with horizontal dotted lines indicating the epidemic threshold ($R_t = 1$). Green and red curves show the estimated $R_t$ values with and without incorporating human mobility, respectively. Shaded areas around the curves denote the 95% credible intervals. The deviations between the two estimates highlights the importance of accounting for human mobility in transmission modeling. Middle row (d–f): Stacked plots illustrate the three sources of daily infections. Orange shows the infections caused by the infected population of that same region. Blue shows the infections that generated within the region by infected incoming population form the other regions. Purple reflects mobility-induced infections acquired outside the region by the outgoing susceptibles. Bottom row (g–i): The proportion of infections from three different sources. The colors are considered same as defined for the Fig 3(d)–3(f). These figures represent the considerable number of mobility driven infections in the disease transmission dynamics.

**Table 1**. **Correlation between one-step increments ($R_t - R_{t-1}$) of true and estimated $R_t$s.**

|  |  | Connected $R_t$ | Disconnected $R_t$ |
|---|---|---|---|
| True $R_t$ | Region 1 | 0.89 | 0.84 |
|  | Region 2 | 0.60 | 0.50 |
|  | Region 3 | 0.80 | 0.69 |

**Table 2**. Correlation between the residuals (disconnected $R_t-$ ground truth) and the proportion of mobility-driven infections.

| | | Residuals (Estimated disconnected $R_t$ - Ground truth) |
|---|---|---|
| Proportion of mobility driven infection | Region 1 | 0.73 |
| | Region 2 | 0.94 |
| | Region 3 | 0.94 |

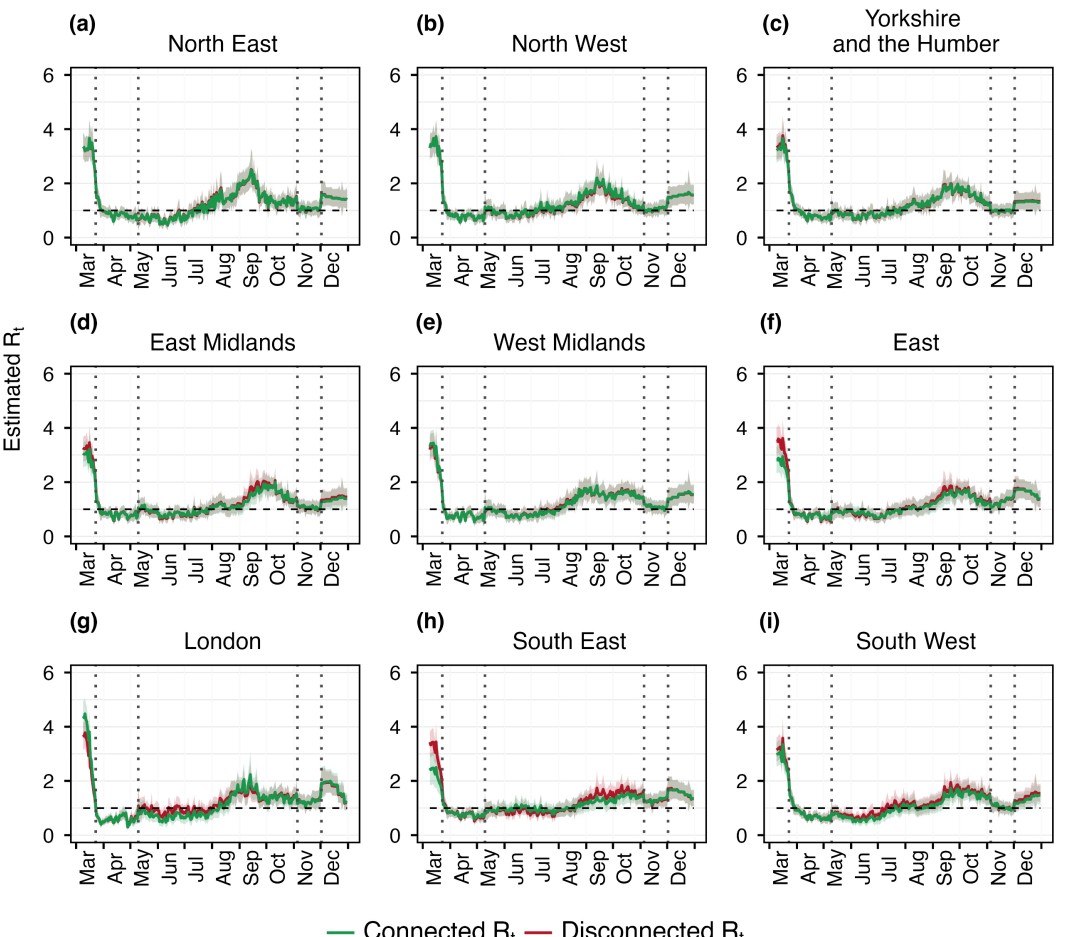

**Fig 4**. **Estimated $R_t$ over different regions of England.** The red curve shows the disconnected $R_t$ which is the reproduction number estimated separately for each region without considering mobility across them. Green curve is the connected $R_t$ that is the estimated reproduction number with the considered mobility matrix showed in S3a Fig. Each curve is shown with its corresponding 95% credible interval. The vertical dotted lines show the first and second lockdown period, and the horizontal black dashed line is the threshold for reproduction number that is $R_t = 1$.

disconnected estimated $R_t$s are nearly identical across all regions. The absolute difference between connected and disconnected $R_t$s are shown in S8 Fig. After an early discrepancy for a few regions (East, London, South East) maximum mean absolute difference reaches 0.25. This suggests that at regional scale, mobility does not have a significant impact on infections after the early transmission phase. Upon examining the mobility matrix (see S3a Fig), it is observed that the fraction of the population commuting between regions is very small relative to the total population of each region. This makes the regions of England treatable as disjoint areas, where the disconnected model is sufficient for $R_t$ estimation at this spatial scale. Transmission at the regional level can be treated as homogeneous spreading within regions, with

minimal influence from inter-regional mobility. This suggests that, when the disease is already established in all regions, mobility can be reasonably excluded from models at such spatial scales. The absolute number and the proportion of daily infections originating from three distinct sources across the 9 regions are shown in S6 and S7 Figs. The Rhat statistics regarding the estimated $R_t$s are shown in S15 Fig.

### 3.3 Findings on higher spatial resolution: LTLAs of North East region

To assess whether the estimates provide any additional or different insights when applied at a higher spatial resolution, where the mobility is considerably greater than the previous case. The high proportion of mobile populations between the LTLAs of North East region has a notable impact on disease transmission, as evidenced by Fig 5. Significant differences

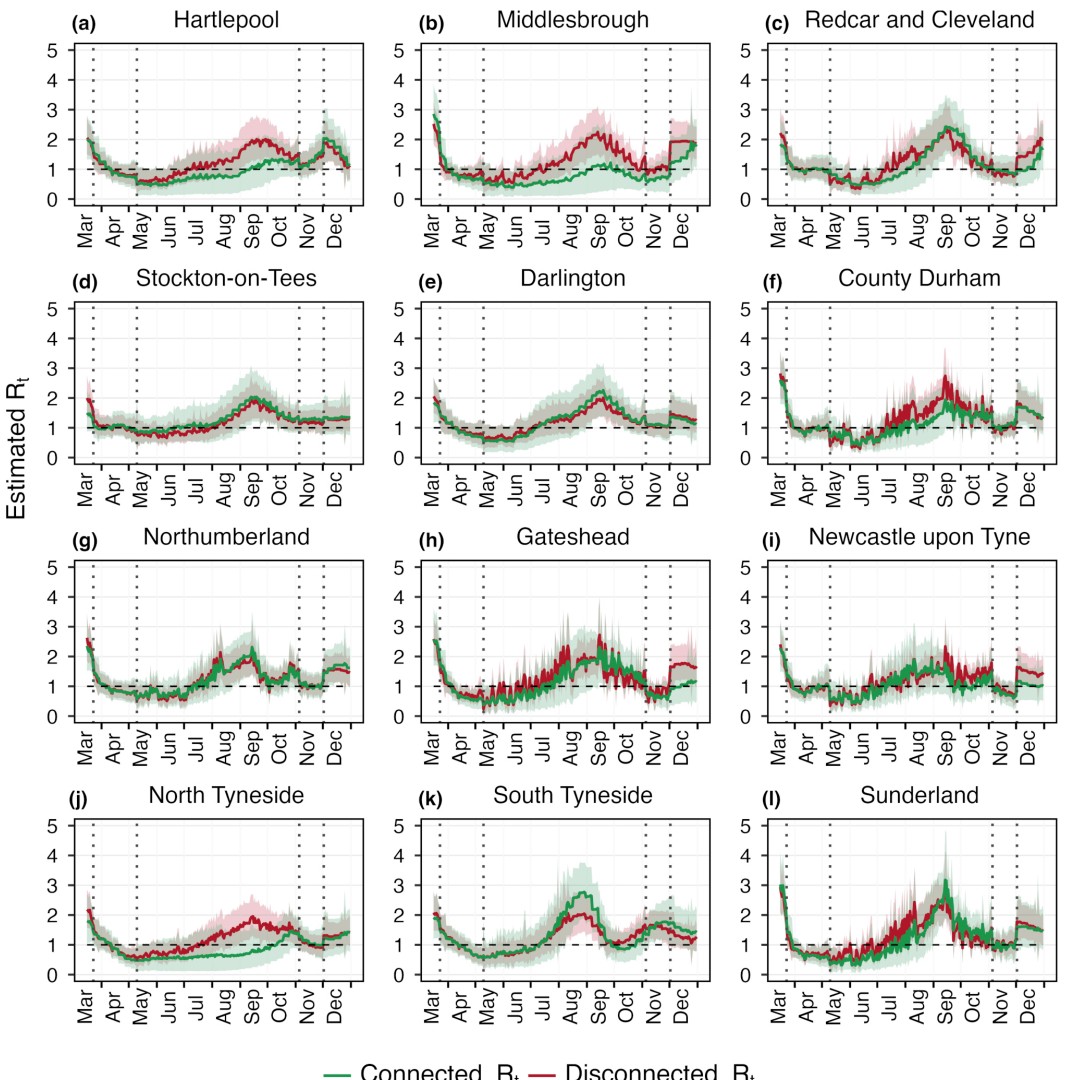

**Fig 5. Estimated $R_t$ over the LTLAs of North East region.** The green and red curve shows the estimation of real-time reproduction number with and without considering mobility respectively. The dotted vertical lines represents the lockdown period and black horizontal line shows the threshold for reproduction number at $R_t = 1$. Red and green bands show the 95% uncertainty interval corresponding to the disconnected and connected $R_t$ estimates. The mobility matrix is shown in S3b Fig.

in the estimated reproduction numbers are observed in LTLAs such as Hartlepool, Middlesbrough, and North and South Tyneside. The absolute difference between connected and disconnected $R_t$s are shown in S11 Fig. In Middlesbrough, the difference reaches 1.09(0.7-1.3) in the second week of September. Similarly, for North Tyneside it reaches 1.15(1.03-1.2). These variations highlight that a considerable number of infections arise from movement between LTLAs, as shown in Fig 6, and the estimated disconnected $R_t$s fail to accurately capture these transmission dynamics, often resulting in under-estimation or overestimation of the disease transmission. This underscores the importance of incorporating intra-regional mobility into models to better understand localized disease transmission patterns.

In Figs 6 and 7, we examine the absolute number and the proportion of daily infections originating from three distinct sources across the 12 LTLAs, providing direct evidence of a substantial portion of infections arising from mobility. In

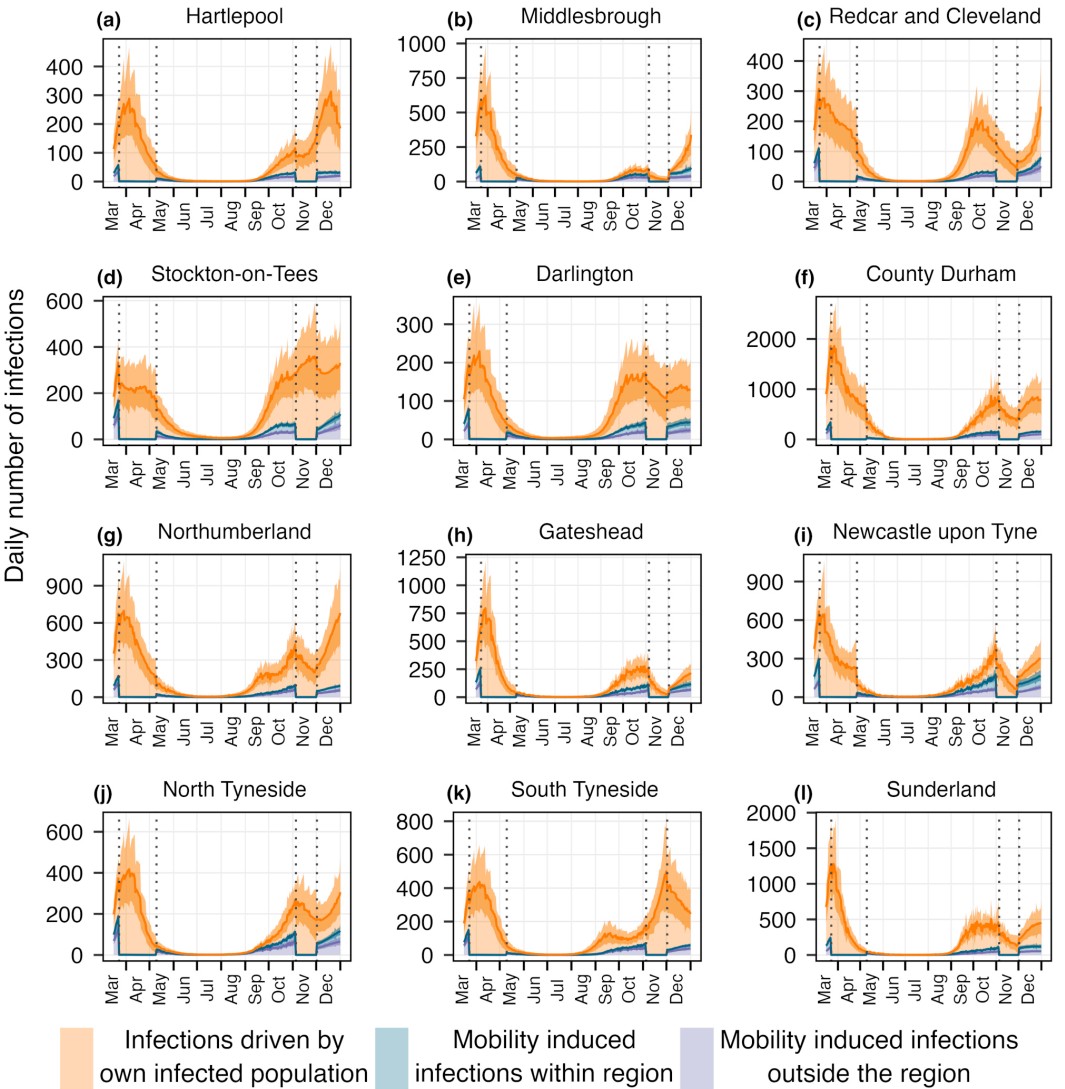

**Fig 6**. **Daily infections from three different sources over the LTLAs of North East region.** Orange color represents the infections occur within the LTLA due to the infected population of that same LTLA. Blue and purple depict the mobility induced infections happen within and outside the LTLA. The dotted vertical lines indicate two lockdown periods while mobility driven infections go down and most of the infections are generated by the local transmission. Here, the lines represent the mean of the estimates and band shows the 95% uncertainty around it.

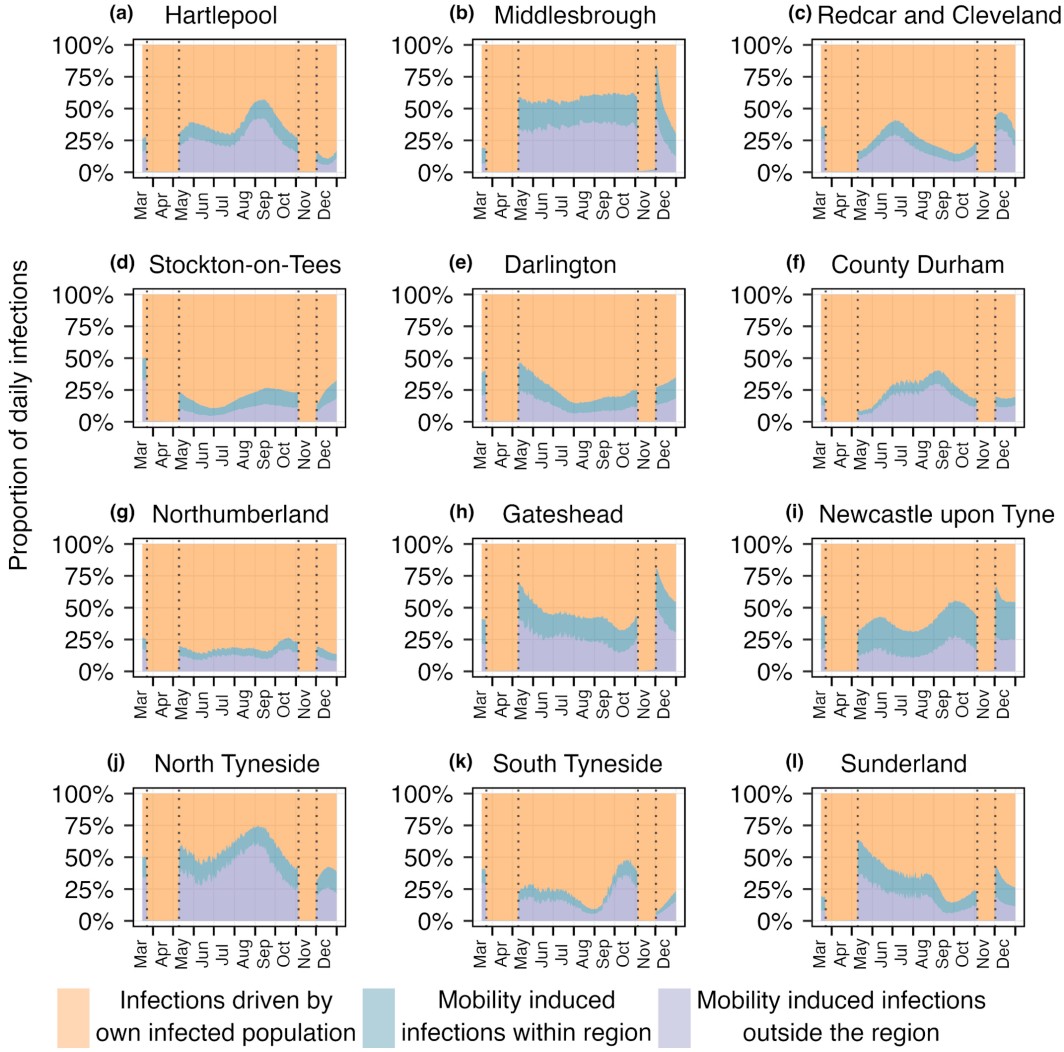

**Fig 7. Proportion of daily infections from three different sources across the LTLAs of the North East region.** Orange, blue, and purple indicate the proportions of daily infections attributed to three distinct sources. The dotted vertical lines represent two lockdown periods, during which nearly all infections resulted from local transmission.

particular, areas such as Hartlepool, Middlesbrough, Newcastle upon Tyne, and North Tyneside show mobility-driven infections contributing to approximately 50% of the total cases, with the proportion rising to as high as 75% in North Tyneside. During the lockdown periods, the reduction in mobility led to a marked decrease in mobility-driven infections, resulting in transmission dynamics being predominantly driven by local spread. These findings underline the crucial role of inter-LTLAs mobility in shaping transmission dynamics and emphasize the importance of accounting for such mobility in public health interventions and resource allocation strategies. Additionally, in S13 and S14 Figs, we present an analysis showing that the main findings are robust to the choice of time scale.

## 4 Discussion

In this study, we developed a spatially extended epidemiological framework that improves the renewal equation based disease transmission model by incorporating human mobility. Unlike approaches that treat mobility as a reductive factor of

behavioral changes within a single region [27,28], we explicitly accounted for incoming and outgoing infected populations, offering a more comprehensive representation of spatial transmission dynamics. The results highlight the substantial role of mobile population in shaping the dynamics of infectious disease transmission and in influencing the estimation of time-varying reproduction numbers ($R_t$). While some recent studies have attempted to integrate mobility flows into transmission models [38], further investigation is needed to address key aspects such as the contribution of different sources of infection and the role of spatial scales. Through applications to both simulated data and real-world SARS-CoV-2 death data in England, our study addresses these limitations and demonstrates that overlooking inter-regional mobility can lead to significant biases in epidemic assessment and incorrect depiction of disease transmission trends.

Using simulated data, we validated our framework by comparing estimated $R_t$ values with known ground truth trajectories. In scenarios with heterogeneous epidemic intensities across regions, we observed that disconnected models (which ignores mobility) often tend to overestimate or underestimate the true $R_t$, as they fail to account for mobility-driven infections that redistribute transmission risk across regions. When mobility is not accounted for, estimated reproduction numbers ($R_t$) of each region tend to closely follow the rise or fall in total infections in that region. In such cases, increasing trend in infections simply lead to higher disconnected $R_t$ estimates, and similarly, when infections decline, disconnected $R_t$ estimates also decrease. It does not account for the infections that happen by the infected population from other regions due to mobility. In contrast, the connected model incorporated a mobility matrix that captured human mobility flow, hence the estimated $R_t$ aligned closely with the ground truth and revealing the key contribution of mobility-driven infections to the observed epidemic patterns. The decomposition of infection sources into local, incoming, and outgoing mobility-driven transmissions provided more insights about the effect of mobility on the transmission process. These results proves the importance of accounting for human movement to improve the precision of epidemic models and real-time surveillance.

We applied this framework to SARS-CoV-2 weekly death data across nine regions of England, our framework revealed that mobility had a limited impact at regional scale, when disease is already established in every region. At this spatial scale (regional level), the proportion of the population commuting between regions was minimal, and transmission occurred primarily within each region. Consequently, disconnected models provided comparable estimates of $R_t$, suggesting that for large administrative units with limited inter-regional mobility, the added complexity of mobility integration may not be essential for real-time epidemic tracking. However, the situation changed markedly at a different spatial resolution. Focusing on the 12 Lower Tier Local Authorities (LTLAs) within the North East region, we observed substantial inter-LTLA mobile population and, correspondingly, a clear difference between connected and disconnected $R_t$ estimates. In certain LTLAs such as North Tyneside and Middlesbrough, mobility-driven infections accounted for up to 75% of total infections at some time points. During lockdown periods, reductions in mobility were reflected in a shift back toward local transmission dominance. These findings underscore that the impact of mobility on transmission and the utility of mobility-informed modeling may depend on the considered spatial scale of the analysis.

Our framework provides a flexible and computationally efficient alternative to individual-based models that require extensive data and computational resources. By incorporating real-world human mobility flows, this model supports a more detailed understanding of how regional transmission trends interact and evolve over time. In the context of spatially heterogeneous epidemics, the impact of population movement on transmission dynamics becomes more pronounced, where differences in reproduction numbers across regions can trigger outbreaks in low-risk areas through population movement. This is especially relevant in real-world contexts, where such disparities are common during the early phases of an epidemic. As the epidemic progresses and spreads more uniformly across regions, local variations may still emerge due to differences in policy implementation, healthcare capacity, demographic profiles, or socio-economic factors. For public health policymakers, these insights are extremely useful. Accurate estimation of spatially connected $R_t$ values can guide targeted interventions, such as localized lockdowns or vaccination prioritization, especially in regions where mobility plays a substantial role in disease spread. Failing to account for these dynamics may result in missed opportunities to suppress transmission or, conversely, in unnecessary restrictions in regions not driving epidemic growth.

A key strength of this framework lies in its generalizability. It can be used for different diseases as the disease-specific epidemiological parameters such as serial interval, incubation period, infection-to-death delay are adapted within the framework. It can be applied to different mobility settings, as an example in this study we considered a high and a low inter-regional mobility settings. The ability to integrate various types of observational data like cases data, hospitalization data, mortality data and even with temporally aggregated data, makes this framework extremely valuable during the early phases of an epidemic or in settings with limited case reporting, and researchers need to rely on the alternative data sources.

A significant challenge when working with mobility-induced transmission models is the limited availability of mobility data. Various datasets serve as proxy for human mobility with its advantages and limitations, at this stage in our case study, we relied on publicly available data to mitigate data constraints. However, these data are often aggregated and may not reflect real-time changes in movement patterns, such as those induced by behavioral shifts or NPIs. Future work could enhance this model by integrating dynamic mobility data, such as mobile phone-based metrics, and extending it to capture other heterogeneities, such as age structure or vaccination coverage. Additionally, specifying a particular mobility threshold that impacts disease transmission is beyond the scope of this work, as we obtained that transmission dynamics are jointly shaped by the mobility and epidemic situations across different regions. Furthermore, the choice of spatial resolution requires careful attention, as at very fine spatial scales, the assumption of homogeneous mixing within each region may be violated.

## 5 Conclusions

This study highlights the importance of incorporating human mobility explicitly within the model to better understand its impact on disease transmission dynamics. The incidence of infections does not solely depend on local outbreaks in a region with commuting populations, instead, our framework demonstrates that commuting populations significantly contribute to the emergence of new infections in connected regions. Consequently, to gain a comprehensive understanding of the epidemic situation in any area, integrating human mobility into the model is essential to mitigate biases that may arise from mobility patterns.

## Supporting information

**S1 Fig. Impact of mobility in a three-region system with identical epidemic conditions.** Each region has an equal population of 1,000,000, the mobility matrices and $R_t$ values are specified in the figure. From left to right, we consider the scenarios with decreasing outgoing mobility and increasing non-commuting population. In the rightmost panel of S1 Fig the regions are disconnected (no inter-regional travel), so the mobility matrix is an identity matrix (all off-diagonal elements are zero). S1d-f Fig shows daily infections by region. Since the epidemic conditions are homogeneous, the incidence trajectories are identical across regions even when outflows differ. However, mobility redistributes the location of infection acquisition. S1g-o Fig shows that, although daily case counts are equal for the three regions, infections occur in different destination regions depending on the mobility pattern.
(TIFF)

**S2 Fig. Impact of mobility in a three-region system with heterogeneous epidemic conditions.** Each region has an equal population of 1,000,000, the mobility matrices and $R_t$ values are specified in the figure. From left to right, we consider the scenarios with decreasing outgoing mobility and increasing non-commuting population. In the rightmost panel of S2 Fig the regions are disconnected (no inter-regional travel), so the mobility matrix is an identity matrix (all off-diagonal elements are zero). S2d–f Fig shows daily infections by region. As the epidemic conditions are heterogeneous over regions, the incidence trajectories are different based on the mobility and local epidemic condition. For example, region 2 is with a controlled situation with $R_t < 1$, however, in S2d–e Fig, there is an outbreak because of the mobility-driven importation, and in the absence of any mobile population it fails to generate any outbreak (See S2f Fig). S2g–o Fig

shows the different sources of infections occur in different destination regions depending on the mobility pattern. These simulations underscore that both human mobility and regional epidemic status jointly shape disease transmission dynamics. Mobility has a significant impact in heterogeneous epidemic scenarios, where differences in transmission intensity across regions can lead to outbreaks in low-risk areas through population movement. This is particularly relevant in real-world settings, where such heterogeneity is common during the early stages of an epidemic. As the epidemic becomes widespread and more uniform across regions, local disparities can appear due to the differences in policy implementation, healthcare capacity, demographic characteristics, or socio-economic conditions.
(TIFF)

**S3 Fig. Mobility matrix.** In this figure, the mobility matrices illustrate the commuting patterns across the regions of England (see S3a Fig) and Lower Tier Local Authorities (LTLAs) within the North East region (see S3b Fig). Each column in the matrix represents the fraction of a region's (or LTLA's) population that commutes to other regions (LTLAs), signifying the outflow from the corresponding region (LTLA) indicated in the figure. On the other hand, the rows depict the inflow to any given region (LTLA) from all other regions (LTLAs). Notably, the diagonal elements (indicated by green box) represent the fraction of the non-commuting population, those who remain within their own region (LTLA). As each column reflects the distribution of a region's (LTLA's) population across all destinations, the sum of each column is equal to one. In S3a Fig, the mobility data for the regions of England reveals that a significant majority of the population does not commute to other regions on a daily basis for work or other purposes. This may be the reason for the lack of observable impact at this broader spatial resolution. In contrast, S3b Fig details a substantial number of commuters among LTLAs within the North East region of England. The data indicates that approximately 15% to 25% of individuals are commuting to other LTLAs, which appears to significantly influence transmission dynamics.
(TIFF)

**S4 Fig. Weekly death data over the nine regions of England.** Vertical lines show the lockdown periods in England.
(TIFF)

**S5 Fig. Model Fitting with the connected and disconnected model for nine regions of England.** The green (red) curve is the fitting with the connected (disconnected) model and 95% credible intervals are shown around the curve. Orange dots are the weekly deaths observed for each region. Dotted vertical lines show the lockdown periods.
(TIFF)

**S6 Fig. Estimated daily infections across nine regions of England from three different sources.** The colored segments illustrate the contributions from each source: orange indicates infections generated by the own infected population within their own region, blue denotes infections acquired locally due to visits of infected individuals from other regions, and purple represents infections contracted outside the region as a result of travel. Vertical lines show the lockdown periods.
(TIFF)

**S7 Fig. Proportion of daily infections for nine regions of.** The proportion of daily infections coming from three different sources and vertical dotted lines represent the lockdown periods.
(TIFF)

**S8 Fig. Absolute residuals between connected and disconnected $R_t$s (shown in Fig 4 in the main text) for 9 regions of England.** Blue dots and red crosses show the mean and median of the absolute differences at each time point. Grey bars show the corresponding 95% credible intervals.
(TIFF)

**S9 Fig. Weekly death data over the twelve LTLAs of North East region.** Vertical lines show the lockdown periods in England.
(TIFF)

**S10 Fig. Model Fitting with the connected and disconnected model for twelve LTLA's of North East region.** The green (red) curve is the fitting with the connected (disconnected) model and 95% credible intervals are shown around the curve. Orange dots are weekly deaths observed for each LTLA. Dotted vertical lines show the lockdown periods.
(TIFF)

**S11 Fig. Absolute residuals between connected and disconnected $R_t$s (shown in Fig 5 in the main text) for 12 LTLAs of North East region of England.** Blue dots and red crosses denote the mean and median of the absolute differences at each time point. Grey bars show the corresponding 95% credible intervals.
(TIFF)

**S12 Fig. Flexibility of the framework in adapting different time-scales.** We consider three regions with populations of 15,00,000, 5,00,000, 10,00,000, respectively. The reproduction numbers are $R_t^{(1)} = 1.1$, $R_t^{(2)} = 1.7$, $R_t^{(3)} = 1.1$, where $R_t^{(i)}$ represents the reproduction number at time $t$ for region $i$. The mobility matrices are considered as follows:

$$C_{day} = \begin{bmatrix} & \text{Region1} & \text{Region 2} & \text{Region 3} \\ \text{Region 1} & 0.7 & 0.07 & 0.21 \\ \text{Region 2} & 0.25 & 0.85 & 0.09 \\ \text{Region 3} & 0.05 & 0.08 & 0.7 \end{bmatrix}$$

$$C_{night} = \begin{bmatrix} & \text{Region1} & \text{Region 2} & \text{Region 3} \\ \text{Region 1} & 1 & 0 & 0 \\ \text{Region 2} & 0 & 1 & 0 \\ \text{Region 3} & 0 & 0 & 1 \end{bmatrix}$$

where, $C_{ij}$ denotes the fraction of population of region $j$ commute to region $i$.

In S12a Fig, we consider a daily time-scale, where the mobility matrix is represented by $C_{day}$ and remains constant throughout the unit time (a full day). In S12b Fig, we consider a finer time scale as half-day intervals. For the first half of the day the mobility matrix is considered as $C_{day}$ and for the second half of the day the mobility matrix is $C_{night}$ to reflect that people typically return to their home regions at night. The generation time distribution is also adjusted accordingly.

Here, $R_t^{(2)} > R_t^{(1)}, R_t^{(3)}$ and therefore, in S12b Fig, when population spends half of the day in their home regions, infections increase in region 2 and decrease in region 1 and region 3 in comparison to S12a Fig.

This shows the flexibility of this framework in adapting different time scales.
(TIFF)

**S13 Fig. Effect of time scale on the estimated $R_t$ over the regions of England.** We consider two time scales: "day" (same as connected $R_t$ as in Fig 4 in the main text), which are represented by the green curves in S13a-i Fig, and "day-night" where unit time is half of a day and represented by the orange curves, and disconnected $R_t$s are shown by the red curves. In the "day-night" case, the first half of a day uses the mobility matrix as shown in S3a Fig, and for the second half we consider the identity matrix to account for the fact that people return to their home region at night. No such difference appears in the results based on the time-scale. We quantify this by calculating absolute residuals $|R_{t,i}^{(day)} - R_{t,i}^{(day-night)}|$ between green and orange curves (see S13a1-i1 Fig). Here, blue dots and purple crosses are the mean and median of the residual distribution at each time point, respectively. The corresponding error bars are represented in light blue color.
(TIFF)

**S14 Fig. Effect of time scale on the estimated $R_t$ for the LTLAs of North East region.** We consider two time scales: "day" (same as connected $R_t$ as in Fig 5 in the main text), which are represented by the green curves in S14a-l Fig, "day-night", where unit time is half of a day and represented by the orange curves, and disconnected $R_t$s are shown by the red

curves. In the "day-night" case, the first half of a day uses the mobility matrix as shown in S3b Fig, and for the second half we consider the identity matrix to account for the fact that people return to their home region at night. We quantify this by calculating absolute residuals $|R_{t,i}^{(day)} - R_{t,i}^{(day-night)}|$ between green and orange curves (see S14a1-l1 Fig). Here, blue dots and purple crosses are the mean and median of the residual distribution at each time point, respectively. The corresponding error bars are represented in light blue color. Therefore, in general results obtained using daily time scale seem robust when comparing them to a potentially more realistic time scale (day-night).
(TIFF)

**S15 Fig. Distribution of Rhat statistics corresponding to the estimated $R_t$s for the 9 regions of England.** Values less than 1.01 indicate MCMC convergence.
(TIFF)

## Acknowledgments

We are thankful to Zhi Ling for helping with computational understanding with STAN. We acknowledge MLGH Network for its role in facilitating scientific exchange that inspired aspects of this research.

## Author contributions

**Conceptualization:** Mousumi Roy, Swapnil Mishra.

**Data curation:** Mousumi Roy, Swapnil Mishra.

**Formal analysis:** Mousumi Roy, Swapnil Mishra.

**Funding acquisition:** Swapnil Mishra.

**Methodology:** Mousumi Roy, Swapnil Mishra.

**Project administration:** Swapnil Mishra.

**Resources:** Swapnil Mishra.

**Software:** Mousumi Roy.

**Supervision:** Swapnil Mishra.

**Validation:** Mousumi Roy, Swapnil Mishra.

**Visualization:** Mousumi Roy, Hannah E. Clapham, Swapnil Mishra.

**Writing – original draft:** Mousumi Roy.

**Writing – review & editing:** Mousumi Roy, Hannah E. Clapham, Swapnil Mishra.

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
