## [Decision Letter · Decision Letter 0]

24 Jun 2025

PCOMPBIOL-D-25-00829

Incorporating human mobility to enhance epidemic response and estimate real-time reproduction numbers

PLOS Computational Biology

Dear Dr. Roy,

Thank you for submitting your manuscript to PLOS Computational Biology. After careful consideration, we feel that it has merit but does not fully meet PLOS Computational Biology's publication criteria as it currently stands. Therefore, we invite you to submit a revised version of the manuscript that addresses the points raised during the review process.

Please submit your revised manuscript within 60 days Aug 24 2025 11:59PM. If you will need more time than this to complete your revisions, please reply to this message or contact the journal office at ploscompbiol@plos.org. Please include the following items when submitting your revised manuscript:

We look forward to receiving your revised manuscript.

Kind regards,

Nicola Perra

Academic Editor

PLOS Computational Biology

Denise Kühnert

Section Editor

PLOS Computational Biology

**Journal Requirements:**

At this stage, the following Authors/Authors require contributions: Mousumi Roy, Hannah E. Clapham, and Swapnil Mishra. Please ensure that the full contributions of each author are acknowledged in the "Add/Edit/Remove Authors" section of our submission form.

Potential Copyright Issues:

i) Figure 1. Please confirm whether you drew the images / clip-art within the figure panels by hand. If you did not draw the images, please provide (a) a link to the source of the images or icons and their license / terms of use; or (b) written permission from the copyright holder to publish the images or icons under our CC BY 4.0 license. Alternatively, you may replace the images with open source alternatives. See these open source resources you may use to replace images / clip-art:

5) We note that your Data Availability Statement is currently as follows: "All relevant data are within the manuscript and its Supporting Information files.". Please confirm at this time whether or not your submission contains all raw data required to replicate the results of your study. Authors must share the “minimal data set” for their submission. PLOS defines the minimal data set to consist of the data required to replicate all study findings reported in the article, as well as related metadata and methods (https://journals.plos.org/plosone/s/data-availability#loc-minimal-data-set-definition). 

2) If any authors received a salary from any of your funders, please state which authors and which funders.

**Reviewers' comments:**

Reviewer's Responses to Questions

**Comments to the Authors:**

**Please note that one of the reviews is uploaded as an attachment.**

Reviewer #1: This work presents a new mathematical approach including mobility between regions to the computation of the net reproduction number. It presents results from a simulation study and from applying the methodology to real data from SARS-CoV-2 in England.

Overall, I found the study interesting. The new equation proposed is quite straightforward and easy to understand, and at the same time it aims to tackle the mobility issue in the computation of the transmissibility in a certain area, which I personally did not see before.

Major suggestion:

However, this study doesn’t take into account the fact that people potentially do not spend all day in another region different from the one where their residence is. An option would be to divide the day in segments in which the matrix “m” changes. Perhaps it would be interesting to see how much this affects the results. Also / Alternatively, it would be nice to justify how your model somehow can account for the fact that the presence of some individuals in other regions only lasts a fraction of the day (week).

Please, find below more detailed comments:

- Abstract: “This analysis also investigates the spatial scalability of the framework, indicating that lower spatial resolution can diminish the effect of inter-regional mobility on the disease transmission, and we conclude that utilizing a finer spatial scale is advantageous on the basis of data availability and computational resources to obtain a better picture of the detailed transmission dynamics.” - Isn’t this the usual result? Is it possible to quantify somehow “how much better / worse” results are with a certain increase / decrease of resolution? Or explain it in terms of the significance of the mobility network?

- Introduction, line 6: SARS-CoV-2.

- Line 7: NPI, to be mentioned first the meaning of the acronym: “non-pharmaceutical interventions (NPIs)”.

- Line 18: models… lead…

- Consider adding parenthesis in equation 2 for easier readability.

- I assumed the “time” in the model is “daily” (discrete nature of the sum), but that could be a bit unrealistic w.r.t. the mobility patterns. I understand that each time step could be anything, in theory? Could it be specified? Human mobility does not usually affect the whole day.

- A bit of repetitions in the text of the methods section: e.g. lines 184-185 explaining the meaning of the diagonal elements of the “m” matrix (people who stay in their region) was already implied before (line 143). Repetition again in lines 206-207.

- Personally, section 2.5 is not fully clear to me. You simulated Rt and not the cases in this experiment? And then the cases are generated from the mobility-inclusive renewal equation. Perhaps an extra line in the prior section specifying Rt will be simulated (assumed), and an extra line in this section explaining the choice to simulate Rt would improve the readability and clarity of the work.

- I miss a short explanation of the choices made in section 2.6 (some distributions, mean choices, etc.). Can be in the Appendix and referenced here.

- For example, line 254: “The 95% credible intervals representing the uncertainty, are shown as the shaded bands…” – shouldn’t all these kind of explanations be restricted to the caption of the figure?

- Results in Figure 3, aren’t they to be expected? The epidemic was simulated using the same equations that now predict Rt better (i.e. connected model). Perhaps results could be presented a bit differently: not in terms of “how much better this model fits w.r.t. to not considering mobility” (expected) but in terms of the risks of computing Rt without mobility and therefore misreading the real transmissibility of the epidemic – pointing out at specific scenarios when this happens (over/under estimation).

- Sentence in rows 280-282 is a clear example of the point above.

- Lines 288-296: again, shouldn’t it be just in the caption of the figure? + Some of those panels described in these lines were already mentioned before. As a reader, I already went through those graphics and now it feels like an unnecessary / misplaced / repetitive explanation.

- Section 3.2: slightly repetitive / long to present a single and clear result: no significant differences observed when mobility affects only a small % of the studied population.

- Section 3.3: Personally, I see this section as one of the key results of this work. Perhaps I missed it, but is the mobility matrix different during lockdown periods? I believe it should change when lockdown was in place. From Figures 6 and 7, it seems mobility was not considered (mij = 0 for i!=j) during lockdown? Please specify it in the text how were lockdowns dealt with (full lockdown, reduced mobility, …).

- Lines 352-355: in the caption.

- The discussion in terms of “finer resolutions” (line 369, for example) shouldn’t it be more accurate saying: when > xx% of the population commutes between studied regions. I understand getting a specific percentage as a threshold might not be possible / easy, but perhaps the text would benefit from a discussion in terms of “presence / absence of significant mobility” and not only in terms of “resolution”. It has been better described in the discussion section already. Also, because, if the “scale is too fine”, the underlying assumption of homogenous mixing inside each region would become problematic. Perhaps add this as a limitation in the discussion.

- Slight “double sense” that came to my mind: finer scale in terms of more regions or in terms of smaller regions? I bet at a scale in which mobility is significant. Just try to be clear on that.

- Lines 452-453: “within” alone (no “into”). // In regions with commuting populations, …

- Lines 461-463: add as limitation in the discussion.

Reviewer #2: the review has been uploaded as an attachment

**Have the authors made all data and (if applicable) computational code underlying the findings in their manuscript fully available?**

Reviewer #1: Yes

Reviewer #2: **No: **References to key data sources are missing

PLOS authors have the option to publish the peer review history of their article (what does this mean?). If published, this will include your full peer review and any attached files.

Reviewer #1: No

Reviewer #2: No

**Figure resubmission:**
---

## [Decision Letter · Decision Letter 1]

5 Oct 2025

PCOMPBIOL-D-25-00829R1

Incorporating human mobility to enhance epidemic response and estimate real-time reproduction numbers

PLOS Computational Biology

Dear Dr. Roy,

Thank you for submitting your manuscript to PLOS Computational Biology. After careful consideration, we feel that it has merit but does not fully meet PLOS Computational Biology's publication criteria as it currently stands. Therefore, we invite you to submit a revised version of the manuscript that addresses the points raised during the review process.

Please submit your revised manuscript within 30 days Dec 05 2025 11:59PM. If you will need more time than this to complete your revisions, please reply to this message or contact the journal office at ploscompbiol@plos.org. Please include the following items when submitting your revised manuscript:

We look forward to receiving your revised manuscript.

Kind regards,

Nicola Perra

Academic Editor

PLOS Computational Biology

Denise Kühnert

Section Editor

PLOS Computational Biology

**Journal Requirements:**

1) We have noticed that you have uploaded Supporting Information files, but you have not included a list of legends. Please add a full list of legends for your Supporting Information files after the references list.

**Reviewers' comments:**

Reviewer's Responses to Questions

**Comments to the Authors:**

Reviewer #1: Thank you for the detailed responses and the throughout revision of my comments. I’m in general satisfied with the changes made on response to my initial comments, and I would suggest it for publication. Only a couple of additional comments:

Fig R3 shows in some cases non-negligible differences on the estimation of Rt. Although is true that, in general, Rt presents a similar trend by considering as time unit either 1 or ½ a day, differences of up to ~0.7 were observed, which can lead to quite different transmission outcomes. Also, finding somewhat different Rt’s when considering different mobility patterns seems reasonable, and somewhat the aim of your work. Perhaps it would be better to say that in the absence of mobility data at a finer scale than “daily”, results seem still robust when comparing them to a potentially “more realistic scenario” (day-night).

Perhaps I am missing something, but section 2.5 is still not clear to me. How are you exactly estimating Rt? Via an MCMC or ABC or…? I miss it to be clearer in this section how the fitting is done, other than the priors and “jumps”. From what I read in page 9, you are using the software CmdStan for model fitting? Try to be organized and clear on the methodological details.

Reviewer #2: I would like to thank the authors for successfully addressing all of my previous points. Methods are now clearly explained and results are supported by quantitative evidence.

I have a couple of (very) minor suggestions on novel additions to the introduction:

1) Lines 20-22: in more advanced compartmental models we can relax, at least partially, the assumption of homogeneous mixing via strategies such as contact matrices. As such, I think it would be more correct to say

"Compartmental mechanistic models [11], in the absence of a spatial component and more advanced mixing assumptions (e.g., contact matrices) may lead to biased estimation due to their assumption of homogeneous mixing, a condition that rarely reflects the real-world mixing pattern."

2) Similarly for the metapopulation model. It is not necessary to assume homogeneous mixing within subpopulations. As such, that claim can be removed or relaxed. I also think the following should be either removed or changed “...approach, it is widely used to study the 29 mechanism behind an outbreak, rather than to evaluate the real time transmission 30 dynamics”. There are notable examples of metapopulation approaches used to study real-time dynamics of infectious diseases, especially at the beginning of emerging outbreaks when it is crucial to estimate importation routes and probabilities, see for example: https://www.science.org/doi/10.1126/science.aba9757, https://pmc.ncbi.nlm.nih.gov/articles/PMC3585792/

**Have the authors made all data and (if applicable) computational code underlying the findings in their manuscript fully available?**

Reviewer #1: Yes

Reviewer #2: Yes

PLOS authors have the option to publish the peer review history of their article (what does this mean?). If published, this will include your full peer review and any attached files.

Reviewer #1: No

Reviewer #2: No

**Figure resubmission:**
---

## [Editor Report · Decision Letter 2]

21 Oct 2025

Dear Dr Roy,

We are pleased to inform you that your manuscript 'Incorporating human mobility to enhance epidemic response and estimate real-time reproduction numbers' has been provisionally accepted for publication in PLOS Computational Biology.

Best regards,

Nicola Perra

Academic Editor

PLOS Computational Biology

Denise Kühnert

Section Editor

PLOS Computational Biology

I am happy to accept the manuscript. The new version of the paper without blue highlights has some formatting issues (references are missing) that should be solved in the next steps

---

## [Editor Report · Acceptance letter]

PCOMPBIOL-D-25-00829R2

Incorporating human mobility to enhance epidemic response and estimate real-time reproduction numbers

Dear Dr Roy,

I am pleased to inform you that your manuscript has been formally accepted for publication in PLOS Computational Biology. Your manuscript is now with our production department and you will be notified of the publication date in due course.

With kind regards,

Judit Kozma
